



***Development of a new Nano-particle sizer equipped with a 12 channel multi-port***
***differential mobility analyzer and multi-condensation particle counters***
**Hong Ku Lee[1], Handol Lee[2] and Kang-Ho Ahn[2],***
*[1]Department of Mechanical Engineering, Hanyang University, Seoul, 04763, R. of Korea.*
*[2]Department of Mechanical Engineering, Hanyang University, Ansan, 15588, R. of Korea.*
**Abstract**
Measuring particle size distributions precisely is an important concern in addressing environmental and human
health-related issues. To measure particle size distribution, a scanning mobility particle sizer (SMPS) is often used.
However, it is difficult to analyze particle size distribution under fast-changing concentration conditions because
the SMPS cannot respond fast enough to reflect current conditions due to the time necessary for voltage scanning.
In this research, we developed a new Nano-particle sizer (NPS), which consists of a multi-port differential
mobility analyzer (MP-DMA) with 12 sampling ports and multi-condensation particle counters (M-CPCs) that
simultaneously measure concentrations of particles classified by the sampling ports. The M-CPC can completely
condense particles larger than 10 nm, and the total particle concentrations measured by each homemade CPC in
the M-CPCs and an electrometer were in agreement up to 20,000 # $cm_{-3}$. For particle classification tests on the
MP-DMA, geometric standard deviations of the size distributions of classified particles were estimated in the
range of 1.035–1.066. We conducted size distribution measurements under steady-state conditions using an
aerosol generator and under unsteady conditions by switching the aerosol supply on/off. The data obtained by the
NPS corresponded closely with the SMPS measurement data for the steady-state particle concentration case. In
addition, the NPS could successfully capture the changes in particle size distribution under fast-changing particle
concentration conditions. For the last, we presented the NPS measurement results of size distributions in common
situation (cooking) as an exemplary real-world application.
**Keywords:** Nano-particle sizer; scanning mobility particle sizer; multi-port differential mobility analyzer;
multi condensation particle counter; real-time particle size distribution; unsteady particle size distribution
• **Corresponding author. Tel: +82-31-417-0601; Fax: +82-31-436-8184**
*E-mail address*: **khahn@hanyang.ac.kr**



## 1 Introduction


There are several methods to measure size distributions of aerosols. Among them, the combination of a
differential mobility analyzer (DMA) and a condensation particle counter (CPC) has been widely used. The
measurement procedure of this technique begins with a voltage applied to the DMA to classify monodisperse
particles in a narrow electrical mobility range, and then the CPC measures the particle number concentration
(Fissan et al., 1983). This is the differential mobility particle sizer (DMPS) method, and by stepping the voltages,
the complete size distribution of aerosols can be obtained. However, generally 10–15 min of the voltage stepping
process are required for accurate estimation of the complete size distribution, making the DMPS unable to respond
accurately if the concentration is changing rapidly. For this reason, the DMPS method has limited applications.
Wang and Flagan (1990) developed a scanning mobility particle sizer (SMPS) to reduce the measurement time.
For the SMPS measurement, the applied voltage is increased (or decreased) continuously, and particles
consecutively classified by a DMA are counted by a CPC. As a result, measurement time can be reduced to less
than two minutes. However, it is still too long to analyze fast-changing particle size distributions. Recently, several
aerosol instrument systems have been developed and studied with the aim of faster measurement. A fast mobility
particle sizer (FMPS) was developed based on a principle similar to the SMPS system, the electrical mobility
analyzer. Instead of a CPC, the FMPS uses multiple electrometers for particle detection, and the system provides
particle size distribution information in real time. The FMPS is generally used for analyzing engine emissions
because the electrometers are not sensitive enough to measure low particle concentrations ($< 10_2$ # cm$_{-3}$). In
addition, current leakage and electrical noise of electrometers sometimes result in less precise measurements. A
new fast integrated mobility spectrometer (FIMS) for real-time measurement of aerosols was developed (Kulkarni
and Wang, 2006). The FIMS detects charged particles based on their different electrical mobilities, which result
in different trajectories. A fast charge-coupled device (CCD) imaging system is employed to capture the locations
of particle trajectories. The FIMS is capable of excellent activation efficiencies for sub-10 nm particles and can
be used to obtain size distributions at sub-second time intervals. Another fast aerosol measurement instrument is
a DMA-train (Stolzenburg et al., 2017). The DMA-train is operated with six DMAs in parallel at a fixed voltage
for particle size distribution measurement with high-time resolution. Therefore, it can be used to observe very fast
aerosol growth, especially in the sub-10 nm range. However, the DMA-train contains six commercial CPCs and
six commercial DMAs, which make the system costly and bulky. Recently, Oberreit et al. (2014) performed
mobility analysis of sub-10 nm particles using an aspirating drift tube ion mobility spectrometer (DT-IMS)



numerically and experimentally. By using the instrument, the electrical mobility of the particles can be estimated
from the time required for the particles to traverse a drift zone. The findings in the paper show that particles
ranging from 2 to 11 nm can be analyzed in less than 5 s. Another instrument for fast measurement is a nucleation
mode aerosol size spectrometer (NMASS) developed by Williamson et al. (2018). The NMASS consists of five
embedded CPCs with different cut-off diameters to measure the particle size distribution between 3 and 60 nm.
To distinguish different diameters, the NMASS requires five different thermal operating conditions for its
condensers.

66       In addition to the above-mentioned instruments, Chen et al. (2007) and Kim et al. (2007) previously developed

a differential mobility analyzer with multiple sampling ports for a fast measurement system. However, the multi-
stage DMA (MDMA) by Chen et al. (2007) has only three sampling ports and needs three CPCs. Furthermore, an
exponentially extended longitudinal length is required to increase the number of sampling ports and accommodate
the wide size range of particles. As a result, the system becomes complicated and expensive. Kim et al. (2007)
developed a DMA with a multi-port system, a substitution for the MDMA system, and it can classify a total of
seven sizes simultaneously. They evaluated the DMA system using monodisperse particles and deduced from the
experiments that increasing the number of sampling ports did not affect the classification efficiency and transfer
functions of the DMA. This was also theoretically supported in research by Giamarelou et al. (2012), in deriving
analytical expressions for estimating the transfer functions and the resolutions of DMAs with multiple sampling
ports. However, there is still a lack of research on a fast measurement system that retains the traditional DMA
function. Therefore, in this study, we developed a new Nano-particle sizer (NPS), consisting of a multi-port DMA
(MP-DMA) and multi-CPCs (M-CPCs), that can perform fast measurement of particle size distributions.
**2 Instrument**
**2.1 Design Concept and Construction of the NPS**

81       The NPS consists of one MP-DMA with 12 ports (Fig. 1(a)) and two M-CPC modules with 12 homemade CPCs

(Fig. 1(c)). The MP-DMA, unlike the common cylindrical DMA with one sampling port (Knutson and Whitby,
1975), has an outer electrode with multiple sampling ports and a truncated cone-shaped inner electrode where a
high voltage is applied. Once the constant voltage is applied, the MP-DMA classifies monodisperse particles
according to their electrical mobility. The dimensions of the entire system are 450 × 300 × 250 mm. The flow
systems and paths for the NPS are depicted in Fig. 1, including the aerosol flowrate ($Q_a$, 0.18 L min-1), sheath


flowrate ($Q_{sh}$, 3.78 L min-1), sampling flowrate ($Q_s$, 0.18 L min-1), and exhaust flowrate ($Q_e$, 1.8 L min-1). Like the
common DMA flow system, $Q_a$ is the same as $Q_s$. The clean sheath flow carries aerosols from the top to the bottom.
Because $Q_s$ continuously flows out through each sampling port, the total flowrate along the classification zone is
reduced.

**2.2 Design Concept of the MP-DMA**

While Chen et al. (2007) employed three sampling ports, and applied an exponentially increasing distance
between neighboring ports to allow a wide size range of particles, the MP-DMA has 12 sampling ports of annular
gaps that circle the entire outer cylinder (electrode) of the MP-DMA, and the ports are placed with a uniform
distance of 2 cm between neighboring ports. To keep the distance short and uniform, the MP-DMA uses an inner
electrode with the increasing diameter along the longitudinal direction. As the diameter of the electrode increases,
the distance between the inner electrode and the outer cylindrical electrode decreases. Accordingly, the electrical
field strength applied to particles increases as they flow to the downstream side. As a result, the MP-DMA can
accommodate a wider size range of particles without excessive extension of the electrode length found in the
common cylindrical electrode.

**2.3 Design Concept of the M-CPC**

Each sampling port in the MP-DMA is directly connected to the inlet of each homemade CPC. Classified
particles are introduced to and measured by the CPC. One M-CPC module consists of six homemade CPCs, and
the NPS has two M-CPC modules (12 CPCs). The module has a unified saturator and condenser block to maintain
uniform temperatures. A common working fluid reservoir is located beneath the saturator block. In this article,
each homemade CPC was denoted as CPC1, CPC2, CPC3, etc., based on their location. CPC1 is closest to the
aerosol inlet and CPC12 is closest to the sheath outlet in the MP-DMA. The reference CPC used in this study is
denoted as TSI-CPC (model 3776, TSI Inc., Shoreview MN, USA).

**Figure 1**

**3 Experimental Setup and Operating Conditions**

**3.1 M-CPC**

In order to evaluate the performance of the M-CPC, the activation efficiency and concentration linearity of each



homemade CPC were obtained from comparison with a reference electrometer. Figure 2(a) is the schematic
diagram of the M-CPC performance test. Using a homemade Collison atomizer, a 0.1 wt% NaCl solution was
atomized, and the aerosols were classified by the first DMA (standard DMA, model 3081, TSI Inc., Shoreview
MN, USA) to generate monodisperse particles which were distributed to the analyzing instruments. In this study,
the operating sheath and aerosol flowrates in the first DMA were 10 L min$_{-1}$ and 1 L min$_{-1}$, respectively. The mode
size and geometric standard deviation of the atomized aerosols were 43.22 nm and 1.65, respectively. The particle
sizes obtained from the atomizer were smaller than 100 nm, thereby minimizing multiple charging effects on the
size-selection (Fig. S1 in Supplementary Material). The concentration of particles was controlled by a diluter
before entering the instruments as shown in Fig. 2(a). To measure the particle number concentration as a reference,
an electrometer (model 6517A, Keithley) with a Faraday cup was used. This is one of the most commonly used
methods for CPC calibration (Liu and Pui, 1974). In this experiment, the sampling flowrate of each CPC was 0.18
L min$_{-1}$, and N-butyl alcohol (Agarwal and Sem, 1980) was used for the working fluid. Temperatures of the
condenser and saturator were controlled to maintain 10 °C and 35 °C, respectively. The M-CPC measured the
number concentration every 1 s, and the response time of the M-CPC is less than 0.3 s. The experimental setup
shown in Fig. 2(a) was used to obtain the results in Fig. 3. For the activation efficiency tests, the tested particle
sizes were 10 nm, 30 nm, 50 nm, 80 nm, and 100 nm. For the concentration linearity test, which is associated with
the detection efficiency of M-CPCs, 50 nm monodisperse particles were used. The tested monodisperse particles
were introduced to the sheath inlet of the MP-DMA with 0 V applied to the inner electrode, and the concentrations
measured by each CPC and the electrometer were compared.
**3.2 MP-DMA**
To evaluate the performance of the MP-DMA, the normalized particle mobility distribution for each port and
penetration efficiency for the MP-DMA were obtained. Figure 2(b) is the schematic diagram of the MP-DMA
performance test. The particle size and concentration were controlled by the first DMA and dilutor, respectively.
The operating conditions of the MP-DMA were 0.18 L min$_{-1}$ for $Q_a$, 0.18 L min$_{-1}$ for $Q_s$, 1.8 L min$_{-1}$ for $Q_e$, and
3.78 L min$_{-1}$ for $Q_{sh}$. The total flowrate ($Q_{sh} + Q_a$) flowing inside the MP-DMA decreases as the flow goes along
the downstream side because each CPC takes 0.18 L min$_{-1}$. Under these flow conditions, the residence time of the
particles flowing from the aerosol inlet to each sampling port inlet is approximately 0.3 s (Port 1) to 3 s (Port 12)
(Lee et al., 2019). The delay due to the residence time inside the MP-DMA was considered when obtaining the
size distributions. In the experiments, the applied voltage on the MP-DMA was fixed, and the stepwise increase


of the voltage on the first DMA was carried out to generate different sizes of monodisperse particles. Their
concentrations were measured by each CPC in the M-CPCs. The upstream concentration of the monodisperse
particles was monitored by the reference TSI-CPC and controlled to approximately 10,000 # cm$_{-3}$ by adjusting
the valve ('B' in Fig. 2(b)) located in the diluter.
With step-wise increase of the voltage on the first DMA, the mobility distributions were obtained from the sets
of measured concentrations as a function of electrical mobility based on the first DMA. The measured
concentrations were normalized by the maximum concentration for each port. The electrical mobility was
normalized by the central mobility for each port, and the results are shown in Fig. 4. In addition, the particle
penetration ratios as a function of port number at voltages of 1000 V and 2000 V are shown in Fig. 5, representing
the maximum ratio between the measured concentration at each CPC and the upstream concentration measured
by the TSI-CPC, which is approximately 10,000 # cm$_{-3}$. The maximum penetration ratio was obtained at the
central electrical mobility for each port. The penetration ratios were used to calibrate the NPS data in the inversion
process.
**3.3 Particle Size Distribution Measurement**
To test the performance of the NPS, the experimental set-up in Fig. 2(c) was used. For particle generation, we
used two types of particles, NaCl and Ag. The NaCl and Ag particles were generated by the homemade Collison
atomizer and evaporation generator (Hwang and Ahn, 2017). The particles were neutralized by a neutralizer, and
the concentration was controlled by a dilutor. The particles were introduced into the TSI-SMPS and NPS. The
TSI-SMPS consists of the standard long DMA (model 3081, TSI Inc., Shoreview MN, USA) and a CPC (model
3775, TSI Inc., Shoreview MN, USA), and the voltage was generated by a high-voltage power supply
(model 205B-10R, Bertan High Voltage, Hicksville NY, USA). The NPS was operated at a constant voltage of
1000 V for size distribution measurements. The performance tests were conducted under steady-state conditions
with constant NaCl and Ag particle concentrations and with changing NaCl particle concentrations during the
transition to the equilibrium state. To provide unsteady particle concentrations, we used the on/off valve at the
aerosol path ('A' in Fig. 2(c)) before the TSI-SMPS and NPS. The total measurement time was 240 s. Two cycles
of the TSI-SMPS measurement were performed consecutively with 120 s scanning time for each cycle, and the
NPS obtained concentration data every 1 s.





**Figure 2**

**3.4 Inversion Process for the NPS Concentration Data**
The raw concentration data measured by the M-CPCs were converted to the real concentrations using an
inversion process considering the multiple charging effect, detection efficiency of the M-CPCs, and penetration
ratio through the MP-DMA. The real concentration of each sampling port was estimated by Eq. (1), and the
multiple charge correction was referred by Hoppel's inversion method (Hoppel, 1978). Variables used in this
inversion process were derived from the experimental results and research of Giamarelou et al. (2012) and
Stolzenburg and McMurry (2008). The correction based on the charge fraction was referred by Wiedensohler's
bipolar charge distribution (Wiedensohler, 1988). For a clear understanding of the variables in Eq. (1), we added
a brief explanation of the experimental method in each result section.
$$\left.\frac{dN}{d\log D_p}\right|_{D_p^*} = \frac{2 \times N_{raw}(D_p^*)_n \times (60/1000)}{f_C(D_p^*)_n \times P(D_p^*)_n \times \eta_{CPC,act}(D_p^*)_n \times \eta_{CPC,det}(N_{raw})_n \times \left\{\log(D_{p,E})_n - \log(D_{p,S})_n\right\}} \tag{1}$$

where $Z_p$ and $D_p$ are the electrical mobility and particle diameter, respectively, $f_c$ is the charge fraction, $P$ is the
penetration ratio, and $\eta_{CPC,act}$ and $\eta_{CPC,det}$ are the activation and detection efficiency of the M-CPC, respectively.
The subscript '$n$' indicates the port number. $D_{p,S}$ and $D_{p,E}$ indicate the particle size range classified by each port.
Because the NPS receives data every 1 s, the raw data with a unit of # s$^{-1}$ were converted to # cm$^{-3}$.
**4 Result and Discussion**
**4.1 Performance of the M-CPC**
Figure 3(a) shows the activation efficiency of the M-CPCs for particles sizes between 10 nm and 100 nm. To
obtain the activation efficiency, monodisperse particles were measured by the electrometer and NPS operated at
0 V as shown in Fig. 2(a). For the NPS measurement, all aerosols were introduced through the sheath flow inlet
only (with a flowrate of 3.96 L min$^{-1}$), so the particle concentrations could be measured by all M-CPCs. The same
flowrate of 3.96 L min$^{-1}$ was introduced to the electrometer, and the measurements were carried out simultaneously.
When comparing the M-CPCs to the electrometer measurements, activation efficiencies of almost 100 % were
obtained for all CPCs for particle sizes down to 10 nm. In this study, we did not find the cut-size of the M-CPC,
but we initially designed the NPS system for detecting particles down to 10 nm.



We also examined the detectable concentration range for the M-CPCs using the experimental setup in Fig. 2(a).
The test was conducted with 50 nm monodisperse particles under different concentration conditions. The
comparison between concentrations obtained by the electrometer and the M-CPCs is shown in Fig. 3(b). The slope
of the graph has a good linearity for concentrations up to 20,000 # cm$_{-3}$, indicating that each homemade CPC can
be used for concentrations up to this value.

201                                    **Figure 3**

**4.2 Performance of the MP-DMA**
The normalized mobility distributions of the MP-DMA's 12 sampling ports were obtained using the experimental
setup in Fig. 2(b), and the results are shown in Fig. 4. The geometric standard deviations for the distributions were
estimated between 1.037 and 1.066, which can be considered a very narrow size classification, indicating that the
resolution of the MP-DMA is fairly good. As mentioned earlier, the total flowrate inside the MP-DMA decreases
as it flows along the downstream side due to the individual sampling ports continuously taking 0.18 L min$_{-1}$. Thus,
the increase in the ratio of $Q_a$ to $Q_{sh}$ results in increasing geometric standard deviation.

209                                    **Figure 4**

Figure 5 shows the penetration ratio of each port in the MP-DMA at voltages of 1000 V and 2000 V. The
penetration ratio is defined as the ratio of the total concentration at the central particle diameter measured by the
NPS to the reference concentration obtained by the TSI-CPC as presented in Fig. 2(b). The penetration ratio of
the MP-DMA ranges from 0.099 to 0.765, and these data were used for calibrating the NPS system to convert the
raw data obtained by the NPS to the reference concentration data. The theoretical resolution of the MP-DMA
decreases from 21 (Port 1) to 10 (Port 12) due to the increasing aerosol-to-sheath flowrate. However, the resolution
of the first DMA (TSI standard DMA) is 10 owing to the ratio between aerosol and sheath flowrate of 1:10.
Therefore, the CPC at Port 1 might count the particles in the narrower size distribution classified by the first DMA,
resulting in a low penetration ratio. Thus, the penetration ratios for all ports were used as correction factors in Eq.
(1) to achieve the same concentration as the reference data measured by the TSI-CPC. Notably, in this experiment,
the reference data are the concentrations of particles classified by the first DMA, and thus the shape of the input
particle size distribution is close to a triangle. Therefore, $N_{raw}/P$ (measured raw concentration divided by the
penetration ratio) represents the area under a triangle. For this reason we multiplied a factor of 2 as shown in Eq.
(1) assuming that a shape of the size distribution of particles entering each port in the NPS is rectangular.





224                                                   **Figure 5**

Figure 6 represents the central particle diameters on each port under different voltage conditions, 1000 V and
2000 V. The classified peak diameter ($D_{peak}$) is the corresponding particle diameter when the concentration of the
classified particles in each port is at its maximum. The classified size range of the NPS is 17–210 nm at 1000 V
and 25–320 nm at 2000 V. The average peak sizes are also listed at the left and right sides of the graph. The range
can be easily adjusted by changing the applied voltage of the NPS. However, there still remains a limitation in the
MP-DMA. There is a blank area between Port 1 and Port 2 where particles with a geometric standard deviation
less than 1.04 (narrow size distribution) and a mode diameter between those of Port 1 and Port 2 are deposited
and will not be detected. However, most real-world aerosol systems have a wide range of size distribution.
Furthermore, the size distribution of aerosols with a geometric standard deviation of 1.04 is rarely seen in actual
applications such as a measurement in ambient air. Therefore, the limitation on the MP-DMA might not result in
critical issues for atmospheric research purposes.

236                                                      **Figure 6**

**4.3 Performance of the NPS**
**4.3.1 Steady-state particle size distribution**
Using the experimental setup in Fig. 2(c), we introduced NaCl or Ag particles to the NPS to measure particle size
distribution, and the results were compared to the TSI-SMPS measurements. The initial concentrations measured
by the NPS were converted to the real concentration based on the inversion process using Eq. (1). Figure 7 shows
particle size distributions estimated by the TSI-SMPS and NPS under steady-state conditions of an aerosol
generator. The data points from the NPS measurements agree with the TSI-SMPS data. Because the NPS has 12
sampling ports and is operated at a fixed voltage, the number of data points is 12. Therefore, to get the complete
size distribution, we fitted the measured data points based on a log-normal distribution. To validate the accuracy
of the fitting method used in this study, we also measured different sets of polydisperse particles (total of ten size
distributions) using the TSI-SMPS and NPS to obtain the mode size and total concentration of each size
distribution, represented in Fig. 8(a) and 8(b). The NPS shows comparable performance to the TSI-SMPS in
measuring particle size and total concentration, and thus, size distribution.

250                                                      **Figure 7**



251                                              **Figure 8**

**4.3.2 Unsteady particle size distribution**
By using the same experimental setup shown in Fig. 2(c), we conducted performance tests for the NPS for
unsteady particle size distribution by employing the on/off valve to introduce or block aerosols to the instruments.
In Fig. 9, the sampling time on the x-axis represents the TSI-SMPS elapsed time after the start of scanning. The
dotted red line in Fig. 9(a) represents the moment we opened the valve ('A' in Fig. 2(c)), indicating introduction
of aerosols 60 s after the beginning of the measurement. In Fig. 9(b), we closed the valve to block the aerosols
180 s after the measurement start. The x-axis and y-axis of the graph for the TSI-SMPS measurement results are
particle diameter and number concentration, respectively. The NPS data is represented in a contour graph with the
sampling time (x-axis) and particle diameter (y-axis). The color indicates the particle number concentration
measured by the NPS.
In Fig. 9(a-1), the concentration data appeared after the valve was opened (60 s after the first scanning process).
However, concentrations for particle sizes below 30–32 nm were not shown during the first scanning process
because the corresponding lower voltages applied for classifying this size range were already scanned when only
the clean air was being measured. In the second scanning process of the TSI-SMPS, the complete size distribution
was obtained. The NPS measurement shows that a few seconds after opening the valve, a rapid increase in particle
concentration for the complete size range was observed, and the size distribution reached steady-state 30–40 s
after the valve was opened. Considering the response time of the NPS is approximately 3.3 s, (sum of the M-CPC
response time of approximately 0.3 s and particle residence time in the MP-DMA, maximum 3 s), the rest of the
delay time might be caused by the time required for concentration stabilization and particle transportation. The
delay was also observed in the TSI-SMPS measurement when comparing the concentration in each channel in Fig.
9(a-1) and 9(a-2). Immediately after opening the aerosol path, the estimated concentration was much lower than
the concentration after reaching steady state in the second scanning cycle. For example, the concentration of 40
nm particles in the first scan was almost 1.5 times lower than the concentration of the same particle size in the
second measurement. During the test for rapid decrease in particle concentration (Fig. 9(b)), the performances of
the TSI-SMPS and NPS are quite distinct as well. After blocking the particle path 180 s after the measurement,
data from the second scanning of the TSI-SMPS show the size distribution for the smaller particles, in a similar
manner to the results in Fig. 9(a), because they were already scanned. However, the size distribution measured by



the NPS completely disappeared after some delay time. Therefore, the NPS can be successfully used for unsteady
particle size distributions to observe changes in concentration.

281                                                    **Figure 9**

NPS measurements under unsteady conditions of rapid-changing particle concentrations were performed for
real-world applications. Figure 10(a) and 10(b) represent particle size distributions measured by the NPS and
SMPS during cooking fish, respectively. The cooking activity was continued for approximately 8 min. The size
distribution obtained by the NPS is shown every 1 s while the SMPS measurement provides one size distribution
every 2 min (total 6 successive measurements). Therefore, the SMPS analysis provides discontinuous size
distribution by time. From the NPS measurements during the cooking activity, particle concentrations varied
significantly. Relatively low particle concentrations were observed approximately 180 s after the beginning of the
activity, and then several peaks were observed until the end of the event. Like these experiments, size distribution
data obtained every 1 s by the NPS can be informative in various applications.

291                                                    **Figure 10**

**5 Conclusion**
In this research, we developed and evaluated the performance of a new Nano-particle sizer (NPS) that measures
particle size distributions under unsteady conditions with changing concentrations. The NPS consists of a
multiport-differential mobility analyzer (MP-DMA) that classifies 12 monodisperse particles of different size and
multi-condensation particle counters (M-CPCs) that count the classified particles. The performances of the MP-
DMA and M-CPC were evaluated by obtaining activation efficiency, detection efficiency, penetration ratio, and
normalized size distribution. The results were used to calibrate the NPS raw data to derive the real particle number
concentration and size distribution. The NPS was compared to the TSI-scanning mobility particle sizer (TSI-
SMPS) for steady-state and unsteady particle concentrations using NaCl and Ag particles. The size distributions
obtained by the NPS under the steady-state condition agreed with the results from the TSI-SMPS. For the unsteady
particle size distribution with fast-changing particle concentration, the NPS was found to be superior to the TSI-
SMPS in terms of measurement speed. However, there remains a needed improvement. During the NPS
measurements, we experienced electrical breakdown when the applied voltage was approximately 4000–5000 V.
Therefore, to improve the NPS system for wider size range classification, further optimization is required. From
the findings in this study, we believe that the NPS can be a promising instrument providing comprehensive



information in applications such as combustion research, diesel emission measurement, and roadside atmospheric
aerosol measurement.



**Acknowledgement**
This work was supported by the research fund of Hanyang University (HY-2019-P).




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





**Captions**
Figure 1. Schematic diagram of the NPS consisting of the MP-DMA including M-CPCs: (a) the geometry of the
MP-DMA and flow paths; (b) the details of the 12th home-made CPC; (c) the M-CPC module.
Figure 2. Schematic diagrams of (a) the M-CPC, (b) MP-DMA and (c) NPS performance tests.
Figure 3. M-CPC performance: (a) Activation efficiencies of 12 home-made CPCs; (b) concentration linearity
between the electrometer and M-CPCs.
Figure 4. Normalized concentrations of the classified particles though each port in the MP-DMA as a function of
normalized electrical mobilities. The $C$ and $C_*$ in the y-axis represent the concentration and the maximum
concentration at each port measured by each home-made CPC, respectively. The data were obtained at the NPS
applied voltage of 1000 V.
Figure 5. Penetration ratio for each port in the MP-DMA.
Figure 6. Peak diameter of the size distribution obtained by using the central mobility range for each port.
Figure 7. Size distributions of the TSI-SMPS and NPS for the constant particle concentrations: (a) Ag particle:
evaporation generator (low temperature), (b) Ag particle: evaporation generator (high temperature), (c) NaCl
particle: Collison atomizer (0.1 wt% NaCl solution). The data were obtained at the NPS applied voltage of 1000
V.
Figure 8. Comparison of (a) mode sizes and (b) total particle number concentrations obtained by the TSI-SMPS
and NPS with NaCl particles. The data were obtained at the NPS applied voltage of 1000 V.
Figure 9. Comparison of the size distributions measured by the TSI-SMPS and NPS for the unsteady particle size
distribution with (a) increasing and (b) decreasing particle concentrations. The tested aerosols were introduced
from 60 seconds after starting measurements: (1) the first TSI-SMPS scanning data; (2) the second TSI-SMPS
scanning data; (3) the NPS data for 240 seconds. The data were obtained at the NPS applied voltage of 1000 V.
Figure 10. Size distributions measured by (a) NPS and (b) SMPS during a cooking activity. The NPS data were
obtained at the applied voltage of 1000 V.



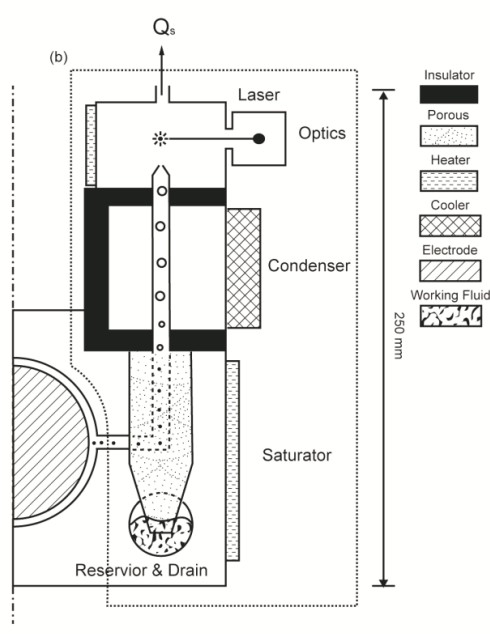


Figure 1



(a)

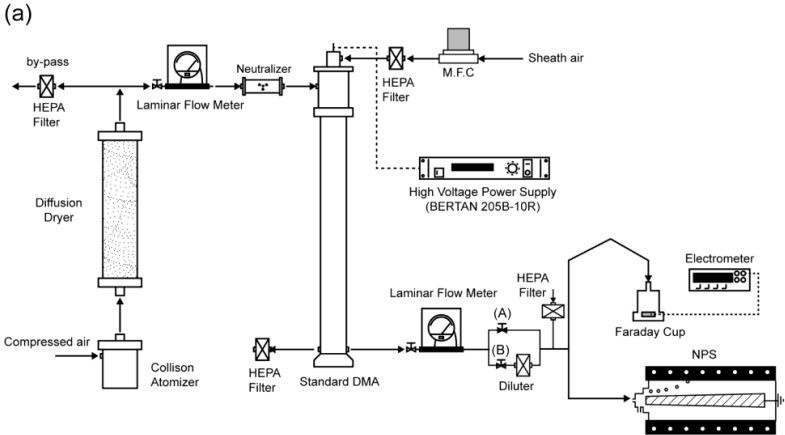

(b)

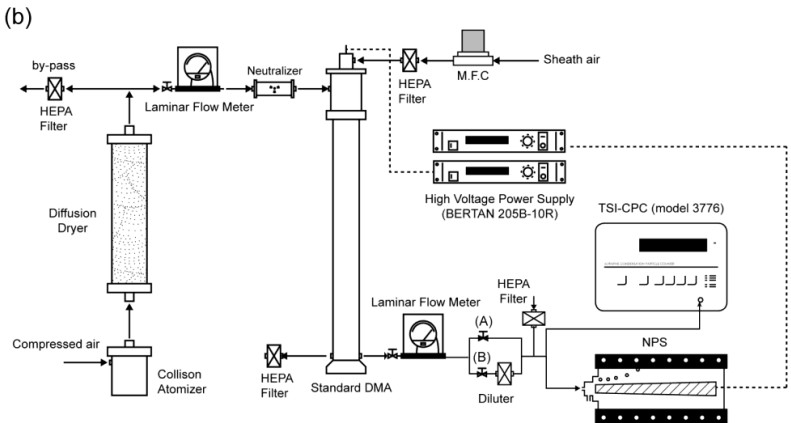

(c)

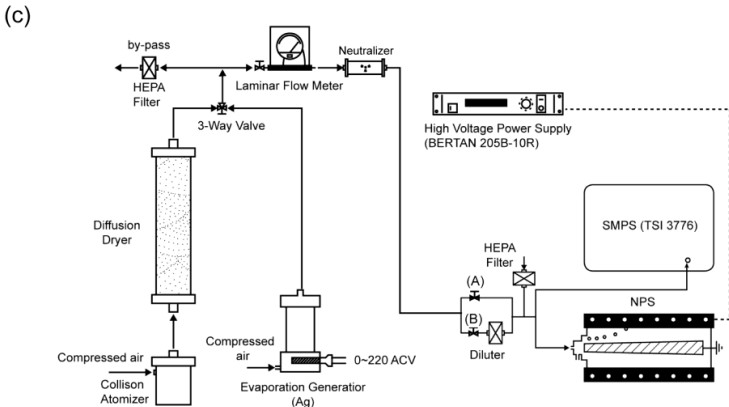


Figure 2





(a)

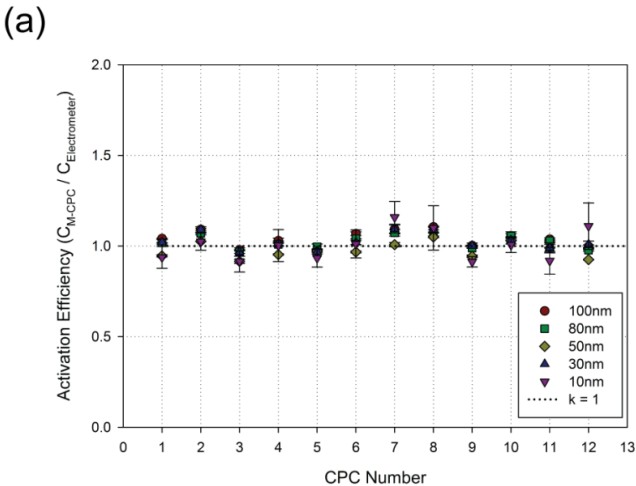

(b)

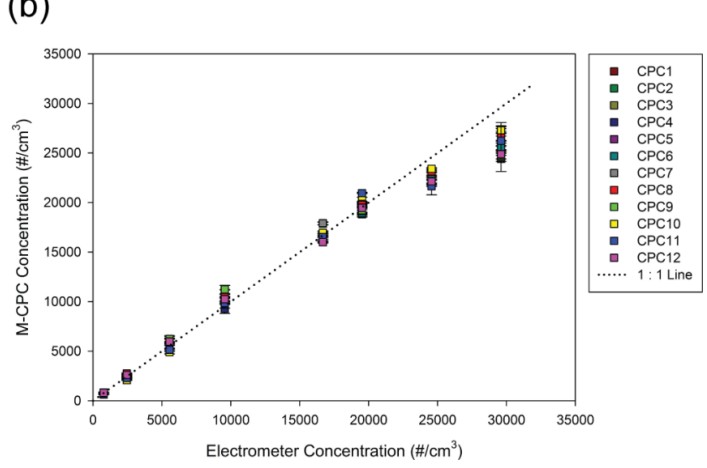


Figure 3



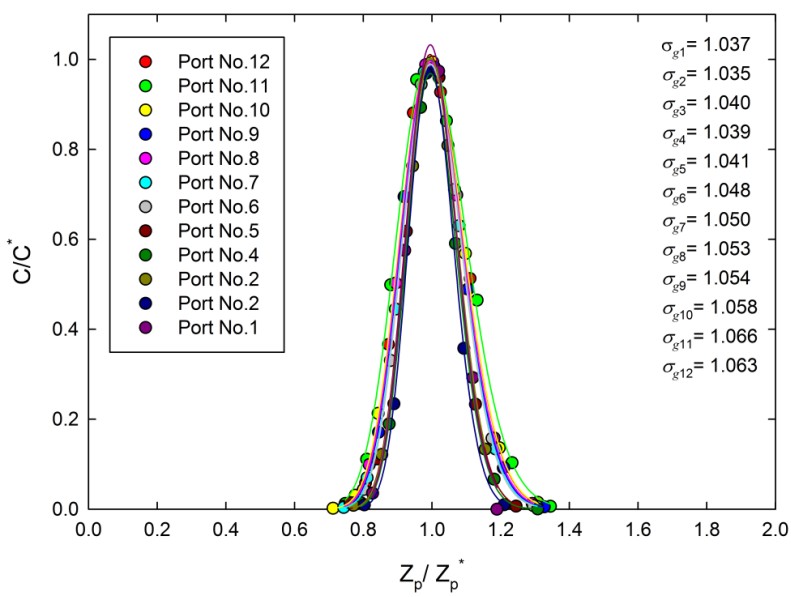


Figure 4





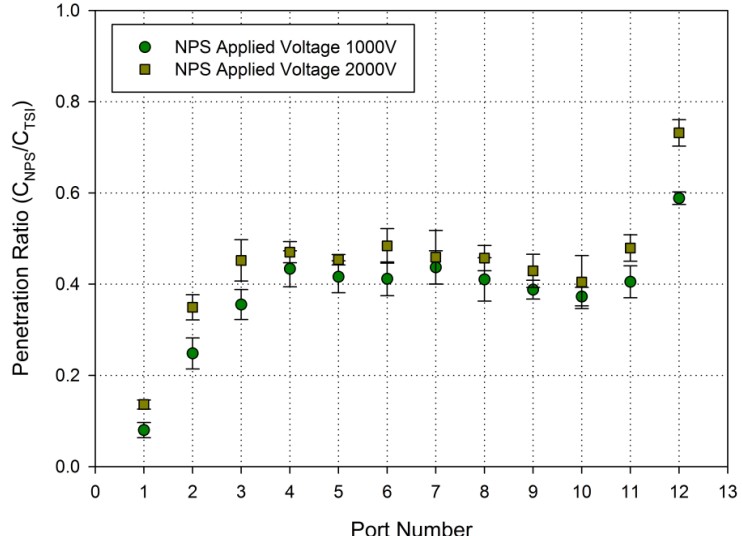


397  Figure 5




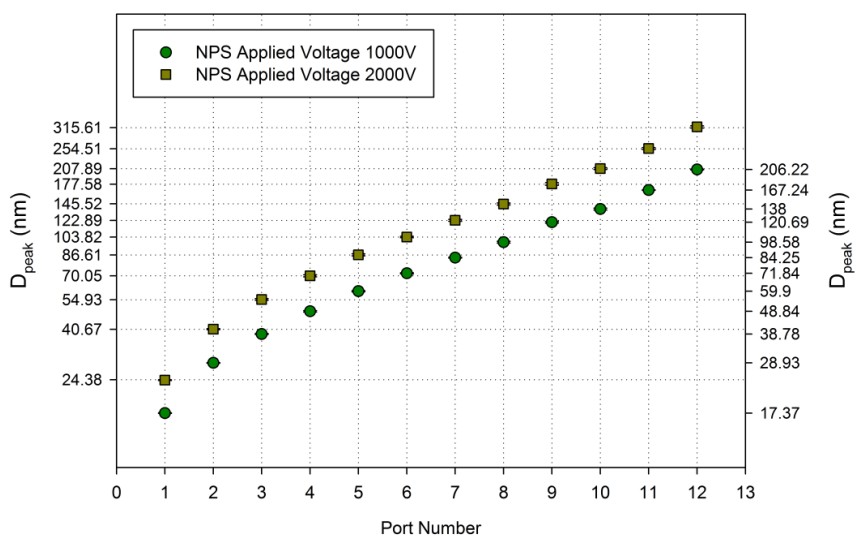


Figure 6



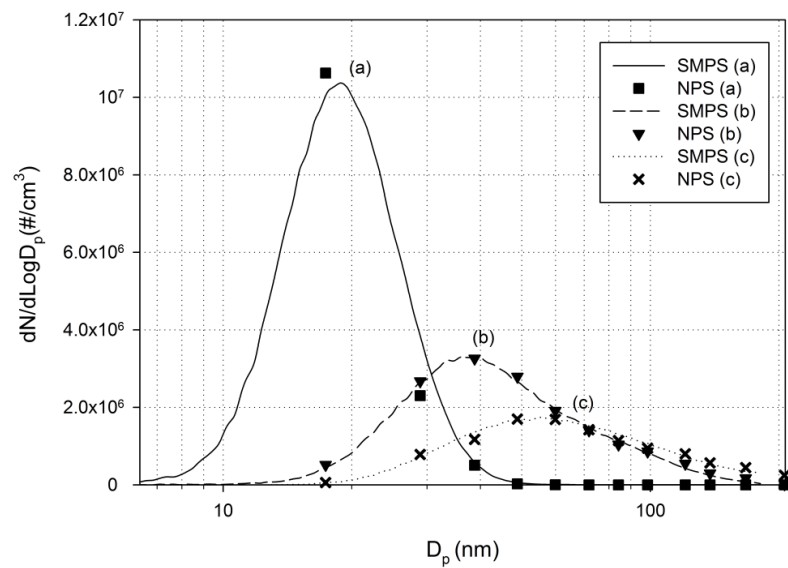


403        Figure 7




(a)

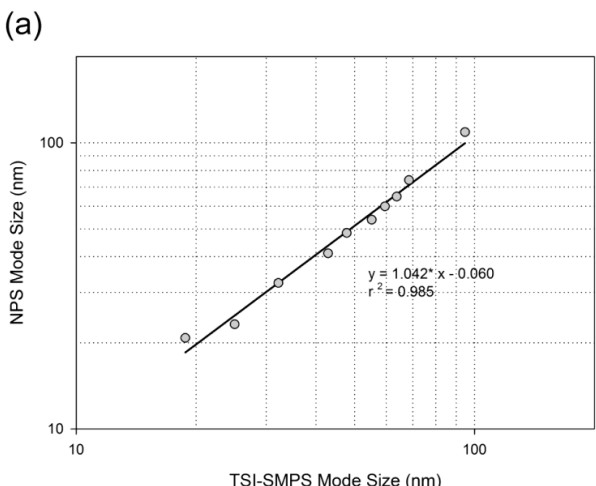

(b)

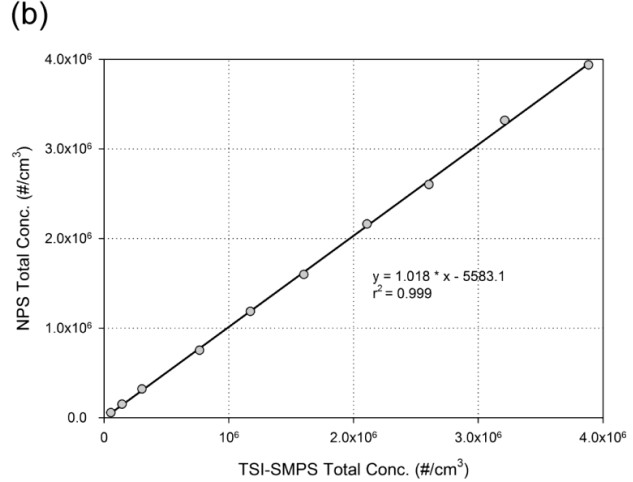


Figure 8





(a)

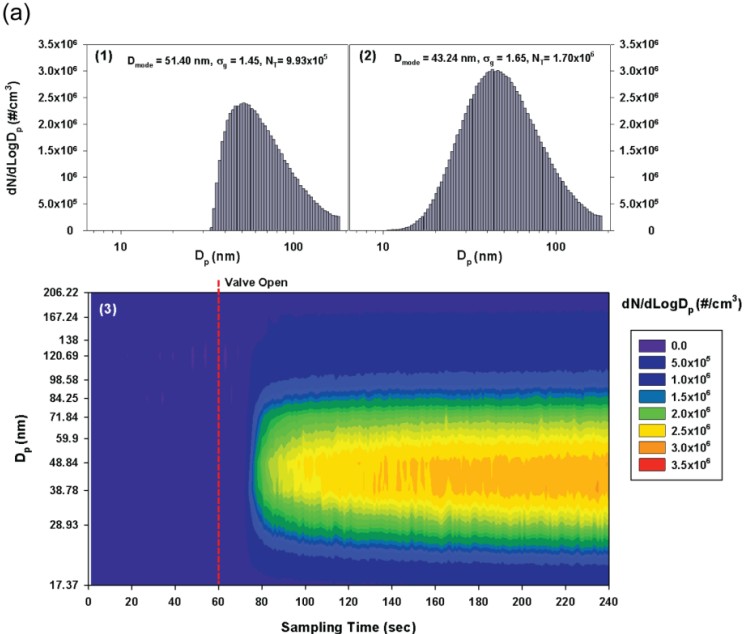

(b)

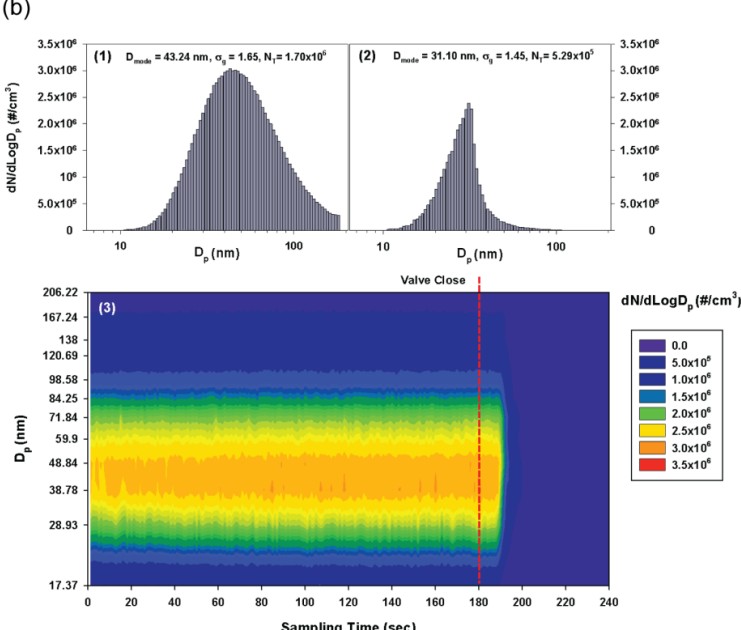


Figure 9





(a)

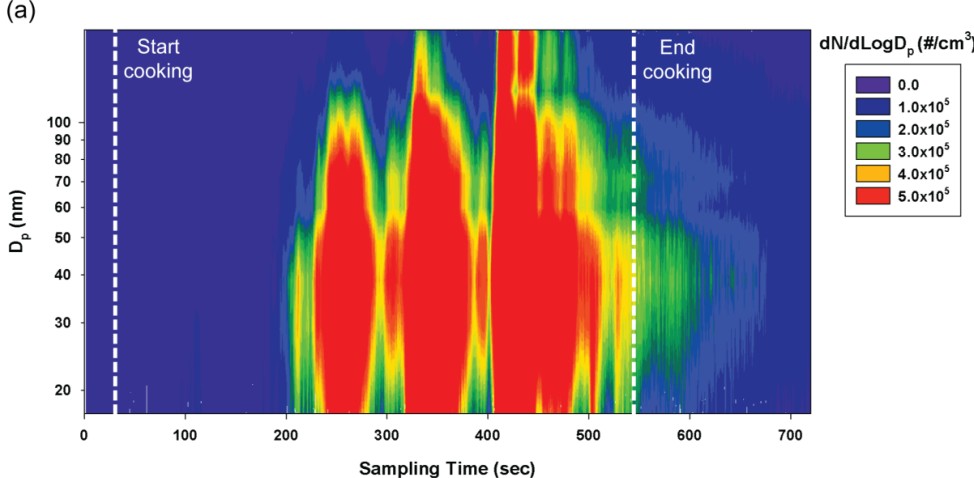

(b)

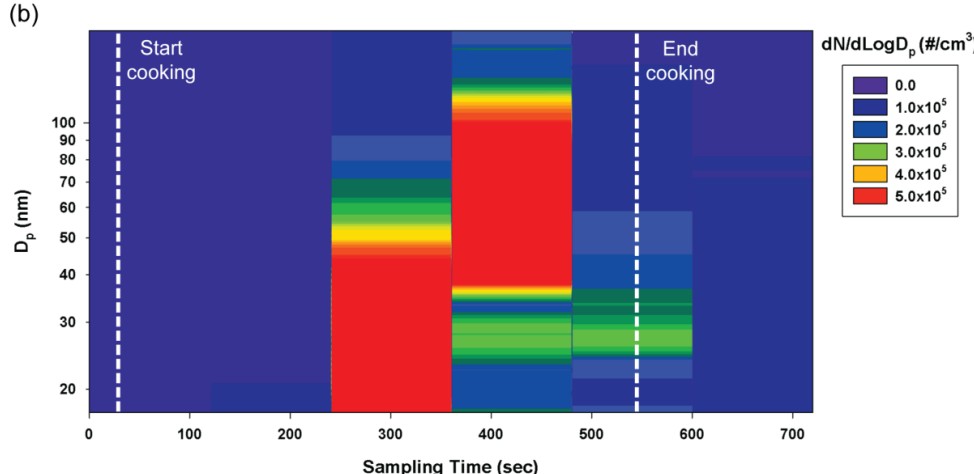


Figure 10