# Peer review of "Development of a new Nano-particle sizer equipped with a 12 channel multi-port"

_Atmospheric Measurement Techniques, 2019_

## Referee Comment (RC1) · Anonymous Referee #1 · 12 Dec 2019

**General comments**

The manuscript presents the experimental work on the development of a nano-particle sizer for measuring a particle size distribution in 1 s time resolution. As a fast measurement system for ambient aerosols has attracted attention, this paper has an originality and deals with important contents. In general, the manuscript shall be considered for the journal publication after some major and minor revisions. Specific comments after reviewing the manuscript are given in the following:

**Major comments**

1. As shown in fig. 1, the inlet of each port seems to be a small hole not an annular ring. Therefore, only a part of introduced particles would be detected by a CPC because particles would be deposited at the wall. So, the particle loss in the MP-DMA might be significant. If particle loss in MP-DMA is high, NPS could not measure low concentration. Then what is minimum measuring concentration of NPS? If the inlet shape of each port is the annular ring, the flow is expected to be deflected. In a typical DMA, the flow deflection is minimized by centering the flow from the annular ring. How did author solve the flow deflection problem in the MP-DMA?

2. TSI-SMPS and NPS showed very good agreement for the particle concentration distribution as shown in fig. 7. Can the NPS detect particles smaller than 17 nm by decreasing NPS voltage? Why did not the authors perform the experiment with the voltage lower than 1000 V?

3. As shown in fig. 9, while the SMPS immediately responded when an aerosol valve was closed or opened., the NPS has 15-20 s response time. Authors explained it with concentration stabilization and particle transportation. However, concentration stabilization might not be the reason because the SMPS responded immediately. Furthermore, particle transportation cannot be the reason if the length of transportation pipe of SMPS and NPS were same. It would be only 3 seconds late even considering the flying time in NPS. Why NPS response time was too late?

**Specific comments**

1. It seems that the difference between concentrations obtained by the M-CPC and electrometer was insignificant in fig. 3b. However, in fig. 5, the difference between the data from the M-CPC and TSI-CPC is large. Authors should explain why the two cases are different so that the reader will not be confused.

2. It should be good to indicate '50 nm monodisperse' in fig. 3b.

3. It might be better to change fig. 6 to a table.

4. It will be better to denote the "valve open" and "valve close" fig. 9 (1) and (2) as well.

5. The minor ticks in the x-axis in fig. 9 (1) and (2) are hard to recognize.

6. Line 85: The NPS seems to be movable. Then, what is the weight of the NPS? Is it hard to move by human hands or not?

7. Line 168: Author mentioned that the maximum flying time of particles inside the NPS is approximately 3 s. Were the NPS data corrected based on the flying time?

8. Line 182: Zp is not presented in Eq. (1), but the description of Zp is shown in the manuscript. Please check the equation.

9. It might be difficult for the readers to understand and compare contour graphs of the NPS and SMPS in fig. 10. It might be better to include in the plot of the obtained mode diameters and concentrations as a function of time.

10. Line 283: The authors should state the positions of the sampling inlets of the NPS and SMPS. The sampling positions for the two instruments should be close to each other for the reliable data comparison. This should be also mentioned in the manuscript.

11. Line 307: In general, particle concentration of diesel emission or roadside atmospheric particles is high, and the authors mentioned in the conclusion that the NPS can be used in these applications. Furthermore, the authors mentioned that the advantage of the NPS is in measuring low concentration of particles in the introduction when compared to the FMPS. The authors need to clearly state the purpose (or applications) of the NPS.

---

## Referee Comment (RC2) · Anonymous Referee #2 · 11 Feb 2020

This manuscript presents a design of a novel differential mobility analyzer with multiple outlets enabling fast parallel measurement of particle size. The manuscript is written mostly in a clear and concise manner presenting the main details of the design of the instrument and tests done to verify its operation. However, some parts of the manuscript explaining the experiments need clarification (see questions below). This manuscript is fit for publications once the questions and comments below have been addressed.

- 1. P. (Page) 4, L. (Line) 87: Suggest changing wording to help the reader to understand the difference between "aerosol flow rate" and "sampling flow rate". Perhaps "sampling flow rate *for each CPC*".
- 2. P. 4, L. 95: As each CPC samples through a single port, how uniform are the sample flows across the circumference of each annulus? One would expect needing multiple ports per annulus to ensure uniformity of flows. Was any CFD modelling done to study the internal flows? Please discuss it.
- 3. P. 4, M-CPC: are there any publications about the M-CPC which could be referenced in this manuscript? If not, then more information about the design and working parameters of the M-CPC should be provided here.
- 4. P. 8, L. 210 and below, also start of P. 9: This paragraph needs elaboration with more explanation provided on how the experiment and data analysis was done. For example, what is meant by "central particle diameter"? How were penetration ratios obtained? Was the TSI SMPS size classification point changed or kept constant? What were the parameters of the aerosol size distribution coming from the SMPS? Was the SMPS data corrected in any way (multiple charging, diffusion losses etc.)? Please add more details.
- 5. Figure 2: Is the SMPS in 2(c) the same as "standard DMA" on 2(a) and 2(b)? If so, state it clearly.
- 6. Figure 3: Is the bias at higher concentrations taken into account in data inversion?
- 7. P. 9, L. 248 and Figure 8: There's a 5500  $cm^{-3}$  bias between the total number concentration measurements from the two instruments, with NPS measuring lower than SMPS. Where does this difference originate from? Is this corrected in data analysis/inversion? Does this mean that the NPS can't measure total particle number concentrations less than 5500  $cm^{-3}$ ? That's a fairly high number for many atmospheric applications. Please discuss.

AMTD
- 8. Figures 7, 8, 9: Were any corrections applied to the SMPS data (multiple charging, diffusion losses etc.)? State this clearly to help the reader make accurate assessments of the results.
- 9. Figure 9: What is meant by first and second scanning data in the figure caption? If these are SMPS scans taken during the measurement, then indicate when they were taken on the NPS color plot. Also, please label the individual plots clearly to indicate from which instrument they are from.

---

## Author Comment (AC1) · 19 Feb 2020

Atmospheric Measurement Techniques (Discussions)

Ref.: Manuscript No. amt-2019-438-RC1

Title: Development of a new Nano-particle sizer equipped with a 12 channel multi-port differential mobility analyzer and multi-condensation particle counters

Anonymous Referee #1, 12 Dec 2019

[Figure]

General comments

The manuscript presents the experimental work on the development of a nano-particle sizer for measuring a particle size distribution in 1 s time resolution. As a fast measurement system for ambient aerosols has attracted attention, this paper has an originality and deals with important contents. In general, the manuscript shall be considered for the journal publication after some major and minor revisions. Specific comments after reviewing the manuscript are given in the following:

Major comments

1. As shown in fig. 1, the inlet of each port seems to be a small hole not an annular ring. Therefore, only a part of introduced particles would be detected by a CPC because particles would be deposited at the wall. So, the particle loss in the MP-DMA might be significant. If particle loss in MP-DMA is high, NPS could not measure low concentration. Then what is minimum measuring concentration of NPS? If the inlet shape of each port is the annular ring, the flow is expected to be deflected. In a typical DMA, the flow deflection is minimized by centering the flow from the annular ring. How did author solve the flow deflection problem in the MP-DMA?

Ans: Thanks for the clarification. In fact, the shape of the sampling ports is annular. We agree that this is not clearly mentioned in the original manuscript. Therefore, we modified the sentence as follows:

Line 94: "While Chen et al. (2007) employed three sampling ports and applied an exponentially increasing distance between neighboring ports to allow a wide size range of particles, the MP-DMA has 12 annular sampling ports that are placed with a uniform distance of 2 cm between neighboring ports."

2. TSI-SMPS and NPS showed very good agreement for the particle concentration distribution as shown in fig. 7. Can the NPS detect particles smaller than 17 nm by decreasing NPS voltage? Why did not the authors perform the experiment with the
voltage lower than 1000 V?

Ans: Thanks for the good comments. We developed the NPS for measuring particle size distribution up to 300 nm particles; therefore, we can utilize the NPS together with an optical particle counter (OPC) for fast ambient particle measurements. As the reviewer mentioned, the NPS might be used at the voltage range under 1000 V to classify smaller particles down to 10 nm. The developed NPS is a prototype, so we are optimizing the flowrate and configuration of the NPS system to characterize smaller particles down to sub-10 nm particles.

3. As shown in fig. 9, while the SMPS immediately responded when an aerosol valve was closed or opened., the NPS has 15-20 s response time. Authors explained it with concentration stabilization and particle transportation. However, concentration stabilization might not be the reason because the SMPS responded immediately. Furthermore, particle transportation cannot be the reason if the length of transportation pipe of SMPS and NPS were same. It would be only 3 seconds late even considering the flying time in NPS. Why NPS response time was too late?

Ans: Thanks for the good comments. In the data processing, we made a mistake. We did not consider the preparation time (about five seconds) of the SMPS before the scanning process. In the experiments, we clicked the start buttons of SMPS and NPS systems simultaneously. The NPS measures size distribution right after the start; however, the SMPS system takes 4-5 seconds before the scanning process, which was previously not considered in the data processing. Therefore, the NPS measures five seconds prior to the SMPS. Based on this, we corrected Fig. 8 in the revised manuscript. Furthermore, we provide a graph on the peak concentration obtained by the NPS as a function of time for two cases (valve open/close) in Figure S2 in Supplementary Material. We found that the particle concentration started to increase 5 s after the valve was opened, and the particle concentration started to decrease 5 s after the valve was closed. Notably, the colored scale might not be enough to capture the small changes in concentration, but we confirmed that the NPS takes approximately 5 s to

respond to the concentration change. With considering the flight time of approximately 3 s in the classification zone in the NPS as well as the residence time before the flow entering the inlet of the NPS. We found that the observed response time until the signal appears for the TSI-SMPS and NPS seems to be reasonable. Again, thanks for pointing it out, so we could find the mistake in the data processing. We added the plot in Supplementary Material.

Line 276: "In Fig. 8(a-1), the concentration data appeared after the valve was opened (60 s after the first scanning process). However, concentrations for particle sizes below 30–32 nm were not shown during the first scanning process because the corresponding lower voltages applied for classifying this size range were already scanned when only the clean air was being measured. In the second scanning process of the TSI-SMPS, the complete size distribution was obtained. The NPS measurement shows that a few seconds after opening the valve, a rapid increase in particle concentration for the complete size range was observed. Specifically, the particle concentration started to increase or decrease approximately 5 s after the valve was opened or closed, respectively (Fig. S2 in Supplementary Material). Considering the response time of the NPS is approximately 3.3 s, (sum of the M-CPC response time of approximately 0.3 s and particle residence time in the MP-DMA, maximum 3 s), the rest of the delay time might be caused by the time required for concentration stabilization and particle transportation. The delay was also observed in the TSI-SMPS. Approximately 2 s after opening the aerosol path (i.e., 1–2 size bins), the concentration started to increase. During the test for rapid decrease in particle concentration (Fig. 8(b)), the performances of the TSI-SMPS and NPS are quite distinct as well. After blocking the particle path 180 s after the measurement, data from the second scanning of the TSI-SMPS show the size distribution for the smaller particles, in a similar manner to the results in Fig. 8(a), because they were already scanned. However, the size distribution measured by the NPS completely disappeared after some delay time. Therefore, the NPS can be successfully used for unsteady particle size distributions to observe changes in concentration."

Specific comments

1. It seems that the difference between concentrations obtained by the M-CPC and electrometer was insignificant in fig. 3b. However, in fig. 5, the difference between the data from the M-CPC and TSI-CPC is large. Authors should explain why the two cases are different so that the reader will not be confused.

Ans: Thanks for the comments. We obtained the activation efficiency (Figure 3) of the M-CPCs with the zero voltage in the MP-DMA (due to the assembled configuration) in order to examine the performance of the M-CPCs (experimental setup: Figure 2a). Therefore, all the particles introduced to the NPS can be measured by the M-CPCs. The results are shown in Figure 3. We modified some sentences for the better understanding of the part.

Line 192: "To obtain the activation efficiency, monodisperse particles were measured by the electrometer and NPS operated at 0 V as shown in Fig. 2(a). For the NPS measurement, all aerosols were introduced through the sheath flow inlet only (with a flowrate of 3.96 L min-1), so the particle concentrations could be measured by all M-CPCs. The same flowrate of 3.96 L min-1 was introduced to the electrometer, and the measurements were carried out simultaneously."

On the contrary, to obtain the penetration efficiency (Figure 5), we operated the NPS at 1000 V and 2000 V (experimental setup: Figure 2b). We used the penetration efficiency as a correction factor to have the same performance as the TSI-CPC. Each port has a different sizing resolution due to the different aerosol-to-sheath flowrate ratios. Therefore, the penetration ratio of the MP-DMA is increasing with the increasing port number (far from the aerosol inlet). The detail of the description for the penetration ratio can be found as follows:

Line 221: "The penetration ratio of the MP-DMA ranges from 0.099 to 0.765, and these data were used for calibrating the NPS system to convert the raw data obtained by the NPS to the reference concentration data. The theoretical resolution of the MP-DMA

decreases from 21 (Port 1) to 10 (Port 12) due to the increasing aerosol-to-sheath flowrate. However, the resolution of the first DMA (TSI standard DMA) is 10 owing to the ratio between aerosol and sheath flowrate of 1:10. Therefore, the CPC at Port 1 might count the particles in the narrower size distribution classified by the first DMA, resulting in a low penetration ratio. Thus, the penetration ratios for all ports were used as correction factors in Eq. (1) to achieve the same concentration as the reference data measured by the TSI-CPC."

2. It should be good to indicate '50 nm monodisperse' in fig. 3b.

Ans: Thanks for the suggestion. We modified Figure 3 as the reviewer recommended.

3. It might be better to change fig. 6 to a table.

Ans: Thanks for the recommendation. We removed Figure 6 and changed it to Table 1.

4. It will be better to denote the "valve open" and "valve close" fig. 9 (1) and (2) as well.

Ans: As the reviewer mentioned, we indicated it in Figure 8 (The figure numbering has been changed). All modified figures are presented in the revised manuscript.

5. The minor ticks in the x-axis in fig. 9 (1) and (2) are hard to recognize.

Ans: We increased the length of the major and minor ticks in Figure 8(1) and 8(2). Please refer to the answer to the previous question (The figure numbering has been changed).

6. Line 85: The NPS seems to be movable. Then, what is the weight of the NPS? Is it hard to move by human hands or not?

Ans: Thanks for the question. The entire system of the NPS is approximately 15 kg. The system can be moved from place to place for sure.

7. Line 168: Author mentioned that the maximum flying time of particles inside the

NPS is approximately 3 s. Were the NPS data corrected based on the flying time?

Ans: Thanks for pointing it out. We already considered the delay from the particle residence time. As the reviewer mentioned the larger particles (latter port) take more time to be classified, so when the size distribution is obtained, the delay factor was considered. This is indicated in the revised manuscript.

Line 144: "The delay due to the residence time inside the MP-DMA was considered when obtaining the size distributions."

8. Line 182: Zp is not presented in Eq. (1), but the description of Zp is shown in the manuscript. Please check the equation.

Ans: Sorry for the confusion. We edited the manuscript as follows:

Line 186: "where Dp is the particle diameter, fc is the charge fraction, P is the penetration ratio, and $\eta$CPC,act and $\eta$CPC,det are the activation and detection efficiency of the M-CPC, respectively."

9. It might be difficult for the readers to understand and compare contour graphs of the NPS and SMPS in fig. 10. It might be better to include in the plot of the obtained mode diameters and concentrations as a function of time.

Ans: We added an additional plot for peak particle concentrations as a function of sampling time, as shown in Figure 9(c).

Line 300: "Figure 9(c) shows the particle concentration at a peak particle size for each measurement of the TSI-SMPS and NPS."

10. Line 283: The authors should state the positions of the sampling inlets of the NPS and SMPS. The sampling positions for the two instruments should be close to each other for the reliable data comparison. This should be also mentioned in the manuscript.

Ans: Thanks for the clarification. As the reviewer mentioned the sampling points for

the SMPS and NPS measurements are close to each other, approximately 10 to 15 cm. Sampling locations are approximately 1 m away from the cooking spot, and the distance from the ground to the sampling port is around 0.8 m. This sampling location, quite close to the cooking spot (1 m), frequently caused sudden changes in concentration. Based on this information we added the comments on this in the revised manuscript.

Line 296: "The sampling location for the TSI-SMPS and NPS measurements is 1 m away from the cooking spot, which caused sudden changes in concentration."

11. Line 307: In general, particle concentration of diesel emission or roadside atmospheric particles is high, and the authors mentioned in the conclusion that the NPS can be used in these applications. Furthermore, the authors mentioned that the advantage of the NPS is in measuring low concentration of particles in the introduction when compared to the FMPS. The authors need to clearly state the purpose (or applications) of the NPS.

Ans: Thanks for the good comments. The advantage of the electrical mobility analyzer system (with a condensation particle counter, CPC) such as SMPS and, in this study, NPS is the wide detection range of concentration (low to high concentrations). Therefore, we take an example of vehicle emission studies in terms of fast-changing concentration condition. As the reviewer mentioned, the statement can be confusing. Therefore, we focused more on the fast-changing concentration conditions and changed the sentence as follows:

Line 320: "From the findings in this study, we believe that the NPS can be a promising instrument providing comprehensive information on fast-changing concentration environments."

[Figure]

---

## Author Response (AR2)

[revised manuscript text omitted]

(b)

[Figure]

(c)

[Figure]

Figure 2

(a)

[Figure]

(b)

[Figure]

Figure 3

[Figure]

Figure 4

[Figure]

Figure 5

[Figure]

Figure 6

(a)

[Figure]

(b)

[Figure]

Figure 7

[Figure]

[Figure]

Figure 8

(a)

[Figure]

(b)

[Figure]

(c)

[Figure]

Figure 9

**Atmospheric Measurement Techniques (Discussions)**

**Ref.: Manuscript No. amt-2019-438-RC1**

**Title: Development of a new Nano-particle sizer equipped with a 12 channel multi-port differential mobility analyzer and multi-condensation particle counters**

Anonymous Referee #1, 12 Dec 2019

**General comments**

**The manuscript presents the experimental work on the development of a nano-particle sizer for measuring a particle size distribution in 1 s time resolution. As a fast measurement system for ambient aerosols has attracted attention, this paper has an originality and deals with important contents. In general, the manuscript shall be considered for the journal publication after some major and minor revisions. Specific comments after reviewing the manuscript are given in the following:**

**Major comments**

**1. As shown in fig. 1, the inlet of each port seems to be a small hole not an annular ring. Therefore, only a part of introduced particles would be detected by a CPC because particles would be deposited at the wall. So, the particle loss in the MP-DMA might be significant. If particle loss in MP-DMA is high, NPS could not measure low concentration. Then what is minimum measuring concentration of NPS? If the inlet shape of each port is the annular ring, the flow is expected to be deflected. In a typical DMA, the flow deflection is minimized by centering the flow from the annular ring. How did author solve the flow deflection problem in the MP-DMA?**

Ans: Thanks for the clarification. In fact, the shape of the sampling ports is annular. We agree that this is not clearly mentioned in the original manuscript. Therefore, we modified the sentence as follows:

Line 94: "While Chen et al. (2007) employed three sampling ports and applied an exponentially increasing distance between neighboring ports to allow a wide size range of particles, the MP-DMA has 12 annular sampling ports that are placed with a uniform distance of 2 cm between neighboring ports."

**2. TSI-SMPS and NPS showed very good agreement for the particle concentration distribution as shown in fig. 7. Can the NPS detect particles smaller than 17 nm by decreasing NPS voltage? Why did not the authors perform the experiment with the voltage lower than 1000 V?**

Ans: Thanks for the good comments. We developed the NPS for measuring particle size distribution up to 300 nm particles; therefore, we can utilize the NPS together with an optical particle counter (OPC) for fast ambient particle measurements. As the reviewer mentioned, the NPS might be used at the voltage range under 1000 V to classify smaller particles down to 10 nm. The developed NPS is a prototype, so we are optimizing the flowrate and configuration of the NPS system to characterize smaller particles down to sub-10 nm particles.

**3. As shown in fig. 9, while the SMPS immediately responded when an aerosol valve was closed or opened., the NPS has 15-20 s response time. Authors explained it with concentration stabilization and particle**

**transportation. However, concentration stabilization might not be the reason because the SMPS responded**
**immediately. Furthermore, particle transportation cannot be the reason if the length of transportation pipe**
**of SMPS and NPS were same. It would be only 3 seconds late even considering the flying time in NPS. Why**
**NPS response time was too late?**

Ans: Thanks for the good comments. In the data processing, we made a mistake. We did not consider the
preparation time (about five seconds) of the SMPS before the scanning process. In the experiments, we clicked
the start buttons of SMPS and NPS systems simultaneously. The NPS measures size distribution right after the
start; however, the SMPS system takes 4-5 seconds before the scanning process, which was previously not
considered in the data processing. Therefore, the NPS measures five seconds prior to the SMPS. Based on this,
we corrected Fig. 8 as follows:

[Figure]

Figure 8. Comparison of the size distributions measured by the TSI-SMPS and NPS for the unsteady particle size
distribution in (a) increasing and (b) decreasing particle concentrations. The tested aerosols were introduced or
blocked 60 s or 180 s after starting measurements, respectively: (1) the first TSI-SMPS scanning data; (2) the
second TSI-SMPS scanning data; (3) the NPS data for 240 s. The data were obtained at the NPS applied voltage
of 1000 V.

Furthermore, we provide a graph below on the peak concentration obtained by the NPS as a function of time for two cases (valve open/close). We found that the particle concentration started to increase 5 s after the valve was opened, and the particle concentration started to decrease 5 s after the valve was closed. Notably, the colored scale might not be enough to capture the small changes in concentration, but we confirmed that the NPS takes approximately 5 s to respond to the concentration change. With considering the flight time of approximately 3 s in the classification zone in the NPS as well as the residence time before the flow entering the inlet of the NPS. We found that the observed response time until the signal appears for the TSI-SMPS and NPS seems to be reasonable. Again, thanks for pointing it out, so we could find the mistake in the data processing. We added the plot in the Supplementary Materials.

[Figure]

Figure S2. Change in particle concentration at the mode diameter as a function of time.

Line 276: "In Fig. 8(a-1), the concentration data appeared after the valve was opened (60 s after the first scanning process). However, concentrations for particle sizes below 30–32 nm were not shown during the first scanning process because the corresponding lower voltages applied for classifying this size range were already scanned when only the clean air was being measured. In the second scanning process of the TSI-SMPS, the complete size distribution was obtained. The NPS measurement shows that a few seconds after opening the valve, a rapid increase in particle concentration for the complete size range was observed. Specifically, the particle concentration started to increase or decrease approximately 5 s after the valve was opened or closed, respectively (Fig. S2 in Supplementary Material). Considering the response time of the NPS is approximately 3.3 s, (sum of the M-CPC response time of approximately 0.3 s and particle residence time in the MP-DMA, maximum 3 s), the rest of the delay time might be caused by the time required for concentration stabilization and particle transportation. The delay was also observed in the TSI-SMPS. Approximately 2 s after opening the aerosol path (i.e., 1–2 size bins), the concentration started to increase. During the test for rapid decrease in particle concentration (Fig. 8(b)), the performances of the TSI-SMPS and NPS are quite distinct as well. After blocking the particle path 180 s after the measurement, data from the second scanning of the TSI-SMPS show the size distribution for the smaller particles, in a similar manner to the results in Fig. 8(a), because they were already scanned. However, the size distribution measured by the NPS completely disappeared after some delay time. Therefore, the NPS can be successfully used for unsteady particle size distributions to observe changes in concentration."

**Specific comments**

**1. It seems that the difference between concentrations obtained by the M-CPC and electrometer was insignificant in fig. 3b. However, in fig. 5, the difference between the data from the M-CPC and TSI-CPC is large. Authors should explain why the two cases are different so that the reader will not be confused.**

Ans: Thanks for the comments. We obtained the activation efficiency (Figure 3) of the M-CPCs with the zero voltage in the MP-DMA (due to the assembled configuration) in order to examine the performance of the M-CPCs (experimental setup: Figure 2a). Therefore, all the particles introduced to the NPS can be measured by the M-CPCs. The results are shown in Figure 3. We modified some sentences for the better understanding of the part.

Line 192: "To obtain the activation efficiency, monodisperse particles were measured by the electrometer and NPS operated at 0 V as shown in Fig. 2(a). For the NPS measurement, all aerosols were introduced through the sheath flow inlet only (with a flowrate of 3.96 L min$^{-1}$), so the particle concentrations could be measured by all M-CPCs. The same flowrate of 3.96 L min$^{-1}$ was introduced to the electrometer, and the measurements were carried out simultaneously."

On the contrary, to obtain the penetration efficiency (Figure 5), we operated the NPS at 1000 V and 2000 V (experimental setup: Figure 2b). We used the penetration efficiency as a correction factor to have the same performance as the TSI-CPC. Each port has a different sizing resolution due to the different aerosol-to-sheath flowrate ratios. Therefore, the penetration ratio of the MP-DMA is increasing with the increasing port number (far from the aerosol inlet). The detail of the description for the penetration ratio can be found as follows:

Line 221: "The penetration ratio of the MP-DMA ranges from 0.099 to 0.765, and these data were used for calibrating the NPS system to convert the raw data obtained by the NPS to the reference concentration data. The theoretical resolution of the MP-DMA decreases from 21 (Port 1) to 10 (Port 12) due to the increasing aerosol-to-sheath flowrate. However, the resolution of the first DMA (TSI standard DMA) is 10 owing to the ratio between aerosol and sheath flowrate of 1:10. Therefore, the CPC at Port 1 might count the particles in the narrower size distribution classified by the first DMA, resulting in a low penetration ratio. Thus, the penetration ratios for all ports were used as correction factors in Eq. (1) to achieve the same concentration as the reference data measured by the TSI-CPC."

**2. It should be good to indicate '50 nm monodisperse' in fig. 3b.**

Ans: Thanks for the suggestion. We modified Figure 3 as the reviewer recommended.

(a)

[Figure]

(b)

[Figure]

Figure 3. M-CPC performance: (a) activation efficiencies of 12 home-made CPCs; (b) concentration linearity
between the electrometer and M-CPCs.

**3. It might be better to change fig. 6 to a table.**

Ans: Thanks for the recommendation. We removed Figure 6 and changed it to Table 1.

Table 1. Mode diameter of the size distribution obtained by using the central mobility range for each port.

| | MP-DMA voltage | Port number | | | | | | | | | | | |
|---|---|---|---|---|---|---|---|---|---|---|---|---|---|
| | | 1 | 2 | 3 | 4 | 5 | 6 | 7 | 8 | 9 | 10 | 11 | 12 |
| Mode diameter [nm] | 1000 V | 17.4 | 28.9 | 38.8 | 48.8 | 59.9 | 71.8 | 84.3 | 98.6 | 120.7 | 138.0 | 167.2 | 206.2 |
| | 2000 V | 24.4 | 40.7 | 54.9 | 70.1 | 86.6 | 103.8 | 122.9 | 145.5 | 177.6 | 207.9 | 254.5 | 315.6 |

**4. It will be better to denote the "valve open" and "valve close" fig. 9 (1) and (2) as well.**

Ans: As the reviewer mentioned, we indicated it in Figure 8 (The figure numbering has been changed).

[Figure]

Figure 8. Comparison of the size distributions measured by the TSI-SMPS and NPS for the unsteady particle size distribution in (a) increasing and (b) decreasing particle concentrations. The tested aerosols were introduced or blocked 60 s or 180 s after starting measurements, respectively: (1) the first TSI-SMPS scanning data; (2) the second TSI-SMPS scanning data; (3) the NPS data for 240 s. The data were obtained at the NPS applied voltage of 1000 V.

**5. The minor ticks in the x-axis in fig. 9 (1) and (2) are hard to recognize.**

Ans: We increased the length of the major and minor ticks in Figure 8(1) and 8(2). Please refer to the answer to
the previous question (The figure numbering has been changed).

**6. Line 85: The NPS seems to be movable. Then, what is the weight of the NPS? Is it hard to move by human
hands or not?**

Ans: Thanks for the question. The entire system of the NPS is approximately 15 kg. The system can be moved
from place to place for sure.

**7. Line 168: Author mentioned that the maximum flying time of particles inside the NPS is approximately**
**3 s. Were the NPS data corrected based on the flying time?**

Ans: Thanks for pointing it out. We already considered the delay from the particle residence time. As the reviewer
mentioned the larger particles (latter port) take more time to be classified, so when the size distribution is obtained,
the delay factor was considered. This is indicated in the revised manuscript.

Line 144: "The delay due to the residence time inside the MP-DMA was considered when obtaining the size
distributions."

**8. Line 182: Zp is not presented in Eq. (1), but the description of Zp is shown in the manuscript. Please**
**check the equation.**

Ans: Sorry for the confusion. We edited the manuscript as follows:

Line 186: "where $D_p$ is the particle diameter, $f_c$ is the charge fraction, $P$ is the penetration ratio, and $\eta_{CPC,act}$ and
$\eta_{CPC,det}$ are the activation and detection efficiency of the M-CPC, respectively."

**9. It might be difficult for the readers to understand and compare contour graphs of the NPS and SMPS in**
**fig. 10. It might be better to include in the plot of the obtained mode diameters and concentrations as a**
**function of time.**

Ans: We added an additional plot for peak particle concentrations as a function of sampling time, as shown in
Figure 9(c).

Line 300: "Figure 9(c) shows the particle concentration at a peak particle size for each measurement of the TSI-
SMPS and NPS."

[Figure]

Figure 9. Size distributions measured by the (a) TSI-SMPS and (b) NPS during a cooking activity and (c) variation
of particle concentration at mode diameters. The NPS data were obtained at the applied voltage of 1000 V.

**10. Line 283: The authors should state the positions of the sampling inlets of the NPS and SMPS. The**
**sampling positions for the two instruments should be close to each other for the reliable data comparison.**
**This should be also mentioned in the manuscript.**

Ans: Thanks for the clarification. As the reviewer mentioned the sampling points for the SMPS and NPS measurements are close to each other, approximately 10 to 15 cm. Sampling locations are approximately 1 m away from the cooking spot, and the distance from the ground to the sampling port is around 0.8 m. This sampling location, quite close to the cooking spot (1 m), frequently caused sudden changes in concentration. Based on this information we added the comments on this in the revised manuscript.

Line 296: "The sampling location for the TSI-SMPS and NPS measurements is 1 m away from the cooking spot, which caused sudden changes in concentration."

**11. Line 307: In general, particle concentration of diesel emission or roadside atmospheric particles is high, and the authors mentioned in the conclusion that the NPS can be used in these applications. Furthermore, the authors mentioned that the advantage of the NPS is in measuring low concentration of particles in the introduction when compared to the FMPS. The authors need to clearly state the purpose (or applications) of the NPS.**

Ans: Thanks for the good comments. The advantage of the electrical mobility analyzer system (with a condensation particle counter, CPC) such as SMPS and, in this study, NPS is the wide detection range of concentration (low to high concentrations). Therefore, we take an example of vehicle emission studies in terms of fast-changing concentration condition. As the reviewer mentioned, the statement can be confusing. Therefore, we focused more on the fast-changing concentration conditions and changed the sentence as follows:

Line 320: "From the findings in this study, we believe that the NPS can be a promising instrument providing comprehensive information on fast-changing concentration environments."

Anonymous Referee #2, 11 Feb 2020

**General comments**

**This manuscript presents a design of a novel differential mobility analyzer with multiple outlets enabling fast parallel measurement of particle size. The manuscript is written mostly in a clear and concise manner presenting the main details of the design of the instrument and tests done to verify its operation. However, some parts of the manuscript explaining the experiments need clarification (see questions below). This manuscript is fit for publications once the questions and comments below have been addressed.**

**1. (Page) 4, L. (Line) 87: Suggest changing wording to help the reader to understand the difference between "aerosol flow rate" and "sampling flow rate". Perhaps "sampling flow rate *for each CPC*".**

Ans: Thanks for the good suggestion. We modified the sentence as follows:

Line 87: "The flow systems and paths for the NPS are depicted in Fig. 1, including the aerosol flowrate ($Q_a$, 0.18 L min$^{-1}$), sheath flowrate ($Q_{sh}$, 3.78 L min$^{-1}$), sampling flowrate for each CPC ($Q_s$, 0.18 L min$^{-1}$), and exhaust flowrate ($Q_e$, 1.8 L min$^{-1}$)."

**2. P. 4, L. 95: As each CPC samples through a single port, how uniform are the sample flows across the circumference of each annulus? One would expect needing multiple ports per annulus to ensure uniformity of flows. Was any CFD modelling done to study the internal flows? Please discuss it.**

Ans: Thanks for the good comments. The "uniform" in the line 95 (in the original manuscript) means that the annular ports are placed with the uniform distance of 2 cm. We agree that this wording might be confusing to readers. Therefore, we deleted the part. As the reviewer mentioned, we recently performed and published the numerical work on the MP-DMA performance, using the computational fluid dynamics (CFD) tool. The numerical simulation focused on flow field and particle transport inside the MP-DMA. The numerically obtained transmission efficiency and resolution agreed well with experimental data. The figure below represents the particle transport (with particle residence time) obtained after flow field simulation. Furthermore, we expect that the flow through each annulus might be quite uniform owing to the small sampling slit (approximately 0.5 mm), which might result in pressure drop and thus uniform flow through the annular slit. It is not easy to observe and evaluate the uniformity of the flow inside the instrument experimentally, but from the consistent results between the experiments and numerical simulations (transmission efficiency and resolution of the MP-DMA), we can assume that flow inside the NPS should be similar to the flow obtained in the simulation, which does not show any uniformity issue. Based on the reviewer's comment, we put some information in the revised manuscript as follows:

Line 96: "The MP-DMA uses an inner electrode with the increasing diameter along the longitudinal direction."

[Figure]

**3. P. 4, M-CPC: are there any publications about the M-CPC which could be referenced in this manuscript? If not, then more information about the design and working parameters of the M-CPC should be provided here.**

Ans: Thanks for pointing it out. In our lab, we developed aerosol instruments including condensation particle counter, optical particle counter, differential mobility analyzer, etc. We have been employing our homemade CPC for investigating atmospheric aerosols. It has the same parts including a saturator, condenser, and optical part. We list the references below that employed our CPC.

(1) Querol, X., Gangoiti, G., Mantilla, E., Alastuey, A., Minguillón, M. C., Amato, F., Reche, C., Viana, M., Moreno, T., Karanasiou, A., Rivas, I., Pérez, N., Ripoll, A., Brines, M., Ealo, M., Pandolfi, M., Lee, H. K., Eun, H. R., Park, Y. H., Escudero, M., Beddows, D., Harrison, R. M., Bertrand, A., Marchand, N., Lyasota, A., Codina, B., Olid, M., Udina, M., Jiménez-Esteve, B., Jiménez-Esteve, B. B., Alonso, L., Millán, M. and Ahn, K. H.: Phenomenology of high-ozone episodes in NE Spain, Atmos. Chem. Phys., 17(4), 2817–2838, doi:10.5194/acp-17-2817-2017, 2017.

(2) Zhu, Y., Wu, Z., Park, Y., Fan, X., Bai, D., Zong, P., Qin, B., Cai, X. and Ahn, K. H.: Measurements of atmospheric aerosol vertical distribution above North China Plain using hexacopter, Sci. Total Environ., 665, 1095–1102, doi:10.1016/j.scitotenv.2019.02.100, 2019.

(3) Minguillón, M. C., Brines, M., Pérez, N., Reche, C., Pandolfi, M., Fonseca, A. S., Amato, F., Alastuey, A., Lyasota, A., Codina, B., Lee, H. K., Eun, H. R., Ahn, K. H. and Querol, X.: New particle formation at ground level and in the vertical column over the Barcelona area, Atmos. Res., 164–165, 118–130, doi:10.1016/j.atmosres.2015.05.003, 2015.

(4) Hwang, I. and Ahn, K. H.: Performance evaluation of conventional type conductive cooling continuous flow compact water-based CPC (Hy-WCPC), J. Aerosol Sci., 113(July), 12–19, doi:10.1016/j.jaerosci.2017.07.007, 2017.

Line 106: "The operating principle of the M-CPC is same as other typical CPCs. Particles are introduced to the saturator (temperature: 35 ℃), and the condensational growth of the particles occurs in the condenser at a temperature of 10 ℃. The condensed particles are detected in the optical part."

**4. P. 8, L. 210 and below, also start of P. 9: This paragraph needs elaboration with more explanation provided on how the experiment and data analysis was done. For example, what is meant by "central**

**particle diameter"? How were penetration ratios obtained? Was the TSI SMPS size classification point changed or kept constant? What were the parameters of the aerosol size distribution coming from the SMPS? Was the SMPS data corrected in any way (multiple charging, diffusion losses etc.)? Please add more details.**

Ans: Thanks for the comments. The answers are presented below.

**4-1) what is meant by "central particle diameter"? How were penetration ratios obtained?**

Ans: The central particle size represents the mode diameter of the classified particles at each port as shown in Table 1 below. The penetration ratio is defined as the ratio between the concentrations of monodisperse particles, generated by the DMA in Fig. 2(b), obtained by each M-CPC and TSI-CPC. For example, we generated 17.4 nm monodisperse particles by using a DMA in Fig. 2(b) and measured the concentrations using the NPS operated at 1000 V and TSI-CPC.

Table 1. Mode diameter of the size distribution obtained by using the central mobility range for each port.

| Mode diameter [nm] | MP-DMA voltage | Port number | | | | | | | | | | | |
|---|---|---|---|---|---|---|---|---|---|---|---|---|---|
| | | 1 | 2 | 3 | 4 | 5 | 6 | 7 | 8 | 9 | 10 | 11 | 12 |
| | 1000 V | 17.4 | 28.9 | 38.8 | 48.8 | 59.9 | 71.8 | 84.3 | 98.6 | 120.7 | 138.0 | 167.2 | 206.2 |
| | 2000 V | 24.4 | 40.7 | 54.9 | 70.1 | 86.6 | 103.8 | 122.9 | 145.5 | 177.6 | 207.9 | 254.5 | 315.6 |

Line 217: "The penetration ratio is defined as the ratio of the total concentration at the central particle diameter (ref. Table 1) measured by the NPS to the reference concentration obtained by the TSI-CPC as presented in Fig. 2(b). For example, monodisperse particles with a mode diameter shown in Table 1 were generated by using a DMA and introduced to the NPS and TSI-CPC to achieve the penetration ratio."

Line 221: "The penetration ratio of the MP-DMA ranges from 0.099 to 0.765, and these data were used for calibrating the NPS system to convert the raw data obtained by the NPS to the reference concentration data. The theoretical resolution of the MP-DMA decreases from 21 (Port 1) to 10 (Port 12) due to the increasing aerosol-to-sheath flowrate. However, the resolution of the first DMA (TSI standard DMA) is 10 owing to the ratio between aerosol and sheath flowrate of 1:10. Therefore, the CPC at Port 1 might count the particles in the narrower size distribution classified by the first DMA, resulting in a low penetration ratio. Thus, the penetration ratios for all ports were used as correction factors in Eq. (1) to achieve the same concentration as the reference data measured by the TSI-CPC."

**4-2) Was the TSI SMPS size classification point changed or kept constant? What were the parameters of the aerosol size distribution coming from the SMPS? Was the SMPS data corrected in any way (multiple charging, diffusion losses etc.)?**

Ans: We employed the TSI-CPC, not TSI-SMPS size classification, for the penetration ratio. As shown in Fig. 2(b) we did not use the scanning-voltage DMA and CPC combination, but we only measured the concentration using the CPC. The DMA in Fig. 2(b) was operated in a fixed voltage mode to generate the monodisperse particles, not scanning voltage mode. The SMPS (scanning voltage mode) was only used for observing the particle size distribution (Fig. 2(c) and from Fig. 7). The SMPS data obtained in this study were corrected based on the multiple charging, charge fraction, and diffusion loss. We used the TSI software to operate the TSI-SMPS system, and this software supports the all the corrections.

Line 261: "For all TSI-SMPS measurements performed in this study, the corrections for the multiple charging and diffusion loss were applied."

**5. Figure 2: Is the SMPS in 2(c) the same as "standard DMA" on 2(a) and 2(b)? If so, state it clearly.**

Ans: Thanks for the suggestion. Yes, the DMA included in the SMPS system is the same as the DMA used in Fig. 2(a) and 2(b). We clearly denoted it in the revised manuscript. Thanks a lot for the clarification.

Line 248: "The TSI-SMPS system consists of the TSI standard DMA and TSI-CPC which were used in Fig. 2(a) or 2(b)."

**6. Figure 3: Is the bias at higher concentrations taken into account in data inversion?**

Ans: That is a good question. Thanks. Yes, we included the correction for the bias at higher concentrations based on the results obtained in this study. However, this high concentration range has never been reached in the real applications such as measuring atmospheric particles. The concentration range shown in Fig 3 represents the performance of the M-CPC. From the experiments, we found the detection limit of the M-CPC by introducing the high concentration of aerosols. However, in the real situation, the M-CPC always measures the concentration of particles classified by the MP-DMA. Therefore, the concentration is usually very low because only small fraction of introduced particles (single positively or negatively charged particles) is detected. As the reviewer pointed it out, we put the correction factor in the NPS system in case of the higher concentration introducing to the M-CPC.

Line 204: "It should be noted that a correction factor was considered in the concentration range higher than 20,000 # cm$^{-3}$. Furthermore, each CPC in the NPS always measures the concentration of particles classified by the MP-DMA; therefore, in real applications such as atmospheric particle measurements, this high concentration after classified by the MP-DMA can be rarely achieved."

**7. P. 9, L. 248 and Figure 8: There's a 5500 cm$^{-3}$ bias between the total number concentration measurements from the two instruments, with NPS measuring lower than SMPS. Where does this difference originate from? Is this corrected in data analysis/inversion? Does this mean that the NPS can't measure total particle number concentrations less than 5500 cm$^{-3}$? That's a fairly high number for many atmospheric applications. Please discuss.**

Ans: As the reviewer mentioned, we observed a 5500 # cm$^{-3}$ bias for the NPS measurement compared to the TSI-SMPS total concentration. We believe that the difference originates from the loss inside the NPS. Due to the low sampling flowrate of 0.18 L min$^{-1}$ for each CPC, there might be additional diffusion loss. We are now optimizing the flowrate and trying to minimize the loss inside the system by increasing the flowrate control system. Therefore, in the future we believe that the bias will be reduced. Thanks a lot for the good comments again. Based on the reviewer's comment, we added a sentence in the revised manuscript.

Line 257: "As shown in Fig. 7(b), we observed the approximately 5500 # cm$^{-3}$ bias in the total concentration for the NPS measurement compared to the TSI-SMPS. We believe that this originates from the particle loss inside the NPS due to the low sampling flowrate for each CPC in the NPS system."

**8. Figures 7, 8, 9: Were any corrections applied to the SMPS data (multiple charging, diffusion losses etc.)? State this clearly to help the reader make accurate assessments of the results.**

Ans: We used the TSI-SMPS system by using the package and the software that the TSI company provides. Therefore, in the system there are options for the multiple charging and diffusion loss corrections. We turned on the corrections when we obtained the data in this study. We stated this in the revised manuscript. Thanks for the good suggestion.

Line 261: "For all TSI-SMPS measurements performed in this study, the corrections for the multiple charging and
diffusion loss were applied."

**9. Figure 9: What is meant by first and second scanning data in the figure caption? If these are SMPS scans**
**taken during the measurement, then indicate when they were taken on the NPS color plot. Also, please label**
**the individual plots clearly to indicate from which instrument they are from.**

Ans: Thanks for the comments. In the experiments, we compared the NPS and TSI-SMPS measurements. Two
cycles of the TSI-SMPS measurement were performed consecutively with 120 s scanning time for each cycle, and
the NPS obtained concentration data every 1 s. Therefore, "first" and "second" in the figure caption represent the
first cycle of 120 s and the second cycle of 120 s for the TSI-SMPS measurements, respectively. The left figure
in Fig. 8(a) (we changed the numbering from Fig. 9 to Fig. 8 in the revision process) represents the TSI-SMPS
data from the first cycle, and the right figure in Fig. 8(b) shows the data from the second cycle. As the reviewer
mentioned, we modified the figure, so it can be more clear to readers.

[Figure]

Fig. 8

**Response to Editor's comments**

Line 10, change "distribution" to "distributions"

Done.

Line 11, change "distribution" to "distributions"

Done.

Line 11, change "concentration" to "concentrations"

Done.

Line 17, I'm not sure this sentence about standard deviations is needed in the abstract. Maybe a statement about the size resolution would be more pertinent.

We agree that the comment on the standard deviations is not necessary, so we deleted the part.

Line 23, change "For the last" to "Finally, we present NPS measurement results. . . "

Done.

Line 54, change "particle trajectories" to "droplets nucleated from these spatially separated particles.

Done.

Line 54, this implies that the FIMS detects sub-10 nm particles; it does not.

Thanks for the comment. We modified the sentence to "The FIMS can be used to obtain size distributions at sub-second time intervals."

Line 85, change "sampling ports" to "sampling ports (annular slits)".

Done.

Line 87, change "with the increasing diameter" to "with increasing diameter"

Done.

Line 109, remove "In this article".

Done.

Line 122, can you really determine the mode diameter to within 0.01 of a nm?

No, it was obtained from the regression line, so we changed the sentence to "The mode size and geometric standard deviation of the atomized aerosols were 43 nm and 1.65, respectively."

Line 215, add "with increasing port number" to the end of the sentence.

Done.

Line 232, are you making a rectangular approximation of full-width at half-max (FWHM), and thus need to apply the factor of 2?

The transfer function is assumed to be triangular for the MP-DMA, so we applied the factor of 2 for obtaining concentrations of entire particles entering sampling slits.

Line 257. I doubt that you actually have a bias of this magnitude--at zero particles you don't count 5500/cm^3, do you? The uncertainty on the intercept must be large enough to explain this. Suggest removing this whole discussion of intercept bias, or confirming that the uncertainty in the fit of the intercept encompasses zero.

Yes, you are right. The part might confuse the readers, so we deleted the discussion on the bias. Thanks for the comment.

Line 267, change to "particle size distributions by employing an on/off valve ('A' in Fig. 2(c)) to introduce. . . ."

Done.

Line 276, change to "60s after the first SMPS scan began."

Done.

Lines 277-278, change to "for particle sizes <32 nm were not recovered from the inversion of this scan because the corresponding voltages were applied to the DMA before the valve was opened, when there were no particles in the sample line."

Done.

Line 279, change "scanning process of the TSI-SMPS" to "scan".

Done.

Line 280, change to "In contrast, the NPS measurement shows a rapid increase in particle concentration for the complete size range soon after the valve was opened."

Done.

Line 285, change "transportation" to "transport". Also suggest removing the entire following 2 sentences, beginning with "The delay was" and ending with "started to increase."

Done.

Line 288, change to "TSI-SMPS and NPS were quite distinct as well. After closing the aerosol valve ~180 s after . . . ."

Done.

Line 289 change to "from the second scan of the TSI-SMPS showed only smaller particles, in a manner similar to the results in Fig. 8(a)."

Done.

Line 290, change to "completely disappeared after some delay time. Therefore," to "quickly approached zero. These tests indicate that the NPS can be. . . ."

Done.

Line 294. Place "Further" before "NPS measurements"

Done.

Line 295, change to "TSI-SMPS and NPS, respectively, during the cooking of fish."

Done.

Line 296. Change "is" to "was"

Done.

Line 300. Change to, "SMPS analysis provides only discontinuous size distributions."

Done.

Line 308. Remove "In this research"

Done.

Line 313 Change "distribution" to "distributions"

Done.

Line 314. Change "the" to "a".

Done.

Line 316. Remove "the" before "steady-state" and before "unsteady".

Done.

Line 317. Add an "s" to "distribution" and "concentration"

Done.

Line 321. Change "can be" to "is"

Done.

Line 321. Add "for" between "instrument" and "providing"

Done.

Line 322. Change to "information on particle size distributions in fast-changing concentration environments."

Done.